# Palliative Care Survey: Awareness, Knowledge and Views of the Styrian Population in Austria

**DOI:** 10.3390/healthcare11192611

**Published:** 2023-09-22

**Authors:** Ulrike Spary-Kainz, Nicole Posch, Muna Paier-Abuzahra, Melanie Lieb, Alexander Avian, Erika Zelko, Andrea Siebenhofer

**Affiliations:** 1Institute of General Practice and Evidence-Based Health Services Research, Medical University of Graz, Neue Stiftingtalstraße 6, 8010 Graz, Austria; ulrike.spary-kainz@medunigraz.at (U.S.-K.); muna.paier-abuzahra@medunigraz.at (M.P.-A.); melaniesoper26@gmail.com (M.L.); andrea.siebenhofer@medunigraz.at (A.S.); 2Institute of Medical Informatics, Statistics and Documentation, Medical University of Graz, Auenbruggerplatz 2, 8036 Graz, Austria; alexander.avian@medunigraz.at; 3Institute of General Practice, Johannes Kepler University Linz, The Life Science Park, 4040 Linz, Austria; erika.zelko@jku.at; 4Institute for General Practice, Goethe University Frankfurt am Main, Theodor-Stern-Kai 7, D-60590 Frankfurt, Germany

**Keywords:** palliative care, awareness of palliative care, information campaign, health literacy

## Abstract

Background: No population-based data on awareness and knowledge of palliative care currently exist in Austria. We therefore conducted a survey to determine the general awareness and knowledge of palliative care in Styria, a federal state in Austria. We also asked participants to imagine what services they would need as a patient or family member, where they themselves would like to receive such services, and what fears they imagined patients with a terminal illness would have. Methods: A descriptive cross-sectional survey consisting of 18 questions that address several aspects of palliative care was carried out in the adult population of Styria, Austria, from October 2019 to March 2020. Results: A total of 419 questionnaires were analyzed, whereby 70.3% of respondents had at least heard of palliative care. Of these, significantly more were female, had a university degree and were aged 50 to 64. The main goal of palliative care was chosen correctly by 67.1% of participants, with the proportion of correct answers increasing in line with education and reaching 82.0% among university graduates. Overall, 73.2% believed that the greatest need of terminally ill persons was a reduction in physical suffering, whereas the greatest perceived need of relatives was the availability of specialist care around the clock. About one-third believed that the greatest fear of palliative patients was that of death, which was chosen significantly more often by men than women. If terminally ill, some 39% of respondents would wish to be looked after at home by professional carers, and women and people that had completed high school chose this answer significantly more often. The most desired service that should be provided to patients and relatives was home pain management at 69.9%, followed by time off for family caregivers at 58.0%. This item was chosen significantly more often by women. Conclusions: To facilitate the care of severely ill patients at home, it would make sense to develop targeted information campaigns. These should also attempt to deliver targeted information to less informed groups of people, such as young, poorly educated men, in order to raise their awareness of the difficulties and challenges of providing care to terminally ill patients and thus increase the acceptance of support options.

## 1. Introduction

Palliative care and palliative medicine have a positive influence on the lives of severely ill patients. The World Health Organization (WHO) defines palliative care as an approach that improves the quality of life of patients and their families who are facing the problems associated with life-threatening illness. It achieves this through the prevention and relief of suffering by means of early identification and the correct assessment and treatment of pain and other problems, whether physical, psychosocial or spiritual [1].

Palliative care improves symptom burden and the health-related quality of life of those affected and increases the likelihood that patients can die at home if they prefer [2]. Although not the primary goal, life can even be prolonged in some cases [3]. In addition to clinical advantages, it also makes financial sense to expand the palliative care network, especially in populations whose life expectancy is increasing [2,3,4]. 

However, expanding the palliative care network requires raising community awareness, eliminating prejudice and teaching people about it. Awareness and knowledge of palliative care vary considerably between countries but also between different groups of people, depending on their age, education and gender. For example, over 80% of respondents to a survey in Northern Ireland said they had heard the term “palliative care”, but most (75%) of them knew little or nothing about it [5], whereas only 32% of respondents to a Turkish survey were aware of it at all [6]. A survey of 3194 persons in the U.S. carried out by Huo et al. showed that fewer than 30% of respondents had any knowledge of palliative care [7]. A Slovenian survey conducted in 2020 showed that women, people of older age and those that had completed higher education were more aware of palliative care [8]. Furthermore, the term “palliative” may be mistakenly associated with hospice care or even euthanasia. A study published in 2013 showed that the term “palliative care” was less familiar to the general population than the term “supportive care”, although they essentially mean the same thing [9]. The term palliative care also carries the stigma of being associated with the very last weeks of life, although early access would be beneficial in terms of quality of life and clinical outcomes [10]. In Styria, a federal state in Austria, institutions specializing in the care of patients with life-limiting illnesses have existed since 1998 [11]. As no population-based awareness data are currently available, we conducted a survey to determine the general awareness and knowledge of palliative care in the general population of Styria, a federal state in Austria. We also asked participants to imagine what services they would need as a patient or family member, where they themselves would like to receive such services, and what fears they would have as a patient.

## 2. Materials and Methods

### 2.1. Study Design

The study is based on a descriptive cross-sectional survey addressing several aspects of palliative care. We therefore surveyed people and asked them to imagine that they were terminally ill patients or family members of such patients. The data were collected as part of an international study performed in Slovenia, Hungary and Croatia. Ethics approval was granted by the Medical University of Graz (EK Nr: 1382/2019 Version 1). 

### 2.2. Participants

Participants included a community-based sample of adults (≥18 years) residing in Styria, a federal state in Austria. Participants were required to understand German, the official language in Austria, and provide their written consent by signing a consent form. 

### 2.3. Recruitment and Data Collection

Two medical practices consecutively recruited the participants, who were subsequently asked to answer an anonymous questionnaire (paper-and-pencil). Names, insurance numbers and other data that could be used to identify any of the respondents were not collected. Data were collected from 9 October 2019 to 25 March 2020. Participants completed the study in one sitting lasting approximately 15 min.

### 2.4. Outcome Measures

We used an existing Slovenian questionnaire [8] that had been used in 1015 Slovenian adults. This survey consists of 18 questions, some multiple choice and some single choice. It was translated from Slovenian into German by a professional translator. To ensure the accuracy of the translated questionnaire, it was proofread by a native Slovenian who is fluent in both German and Slovene and works in the medical field. The questionnaire was linguistically adapted to take into account the differences between the Austrian and Slovenian healthcare and education systems. We were therefore in regular contact with the Slovenian research team during the translation process to ensure that the survey was not only valid and suitable for international comparisons but also appropriate for use in Austria. The German questionnaire was tested using the Cognitive Debriefing method [12]. This is a process by which an instrument is actively tested among representatives of the target population and target language group to determine whether respondents understand the questionnaire properly.

The questionnaire asked about awareness and knowledge of palliative care, and it further asked participants to put themselves in the position of terminally ill patients and their relatives and to provide their views on the services they thought such patients or the family members of such patients should have, where they themselves would like to receive such services (at home, in a hospice, etc.) and what fears they would have if they were a patient. The full questionnaire can be found in the Appendix A and Appendix A.

### 2.5. Statistical Analyses

SPSS 28 (IBM Corp., Armonk, NY, USA) was used for data analysis; *p* < 0.05 was considered as significant. Missing data were not imputed. Data are presented as absolute or relative frequencies. Differences between groups (gender, age groups, educational levels) were analyzed using the Chi-square test or Fisher’s exact test, as appropriate. For this analysis, age was grouped into 18–34 years, 35–49 years, 50–64 years, ≥65 years, and educational level was grouped into EL1: compulsory school, EL2: apprenticeship, EL3: trade school, EL4: high school, EL5: university.

## 3. Results

### 3.1. Description of the Sample

A total of 419 questionnaires were analyzed. A further 37 questionnaires were not taken into consideration because they were incomplete.

The age of the respondents ranged from 18 to 93 years. Overall, 58% were female, 8.3% had completed compulsory schooling (EL1), 28.2% had completed an apprenticeship (EL2), 16.7% had been trained at a technical college or trade school (EL3), 18.1% had finished high school (EL4) and 28.6% had graduated from a university or university of applied sciences (EL5). Overall, 303 lived in a rural area (72.3%) and 116 in an urban area (27.7%). 

### 3.2. Awareness and Knowledge of Palliative Medicine 

Of the 419 respondents, 70.3% (n = 295) had at least heard of palliative medicine in Styria. Of these, 11.2% reported knowing a fair amount about it and 3.3% reported having extensive knowledge. More of the participants that had heard of palliative medicine were female (78.3%, *p* < 0.001), had a university degree (74.2%, *p =* 0.033) and were aged 50 to 64 (79.6% *p =* 0.003). Only the 295 (70.3%) that had at least heard of palliative care were asked what its main aim was. The right answer, which was to “improve quality of life”, was chosen by 67.1% of them (wrong answers: 25.4%; did not know: 7.5%). While the responses did not differ according to sex (*p =* 0.503) and age group (*p =* 0.227), the proportion of correct answers increased in line with education, reaching 82.0% in graduates from universities of applied sciences or universities (*p* = 0.003).

### 3.3. Survey Participants’ Views on the Greatest Needs of Palliative Care Patients and Their Relatives

Overall, 73.2% believed that the greatest need of terminally ill persons at the end of their lives was a *reduction in physical suffering*, while 9.5% said it was *home nursing care*, 6.4% *medical specialist care*, 4.4% *psychological care*, 3.4% *caregiver support* and 1.4% *spiritual support*. Views on the greatest needs of terminally ill persons depended on the respondent’s age. Younger respondents more often believed that *psychological care* was the greatest need of terminally ill persons (age < 35 years: 10.1%, 35–49 years: 4.0%, 50–64 years: 3.7%, ≥65 years: 0.0%; *p =* 0.031). No other significant correlations existed between the provided responses and the variables of age, education and gender. For more details, see Figure 1 and Table A1 in Appendix C.

*Around–the–clock availability of specialist care* was the most common response to the question on the greatest needs of relatives (35.3%), followed by *home nursing* care (26.1%) and *psychological care* (16.3%). A perceived need for home nursing care was stronger in males than females (f: 22.0%, m: 33.7%; *p* = 0.029), while age influenced a perceived need for *around*–*the*–*clock availability of specialist care* (<35 years: 22.9%, 35–49 years: 33.3%, 50–64 years: 36.6%, ≥65 years: 48.5%; *p* = 0.017) and *psychological care* (<35 years: 27.1%, 35–49 years: 20.0%, 50–64 years: 14.6%, ≥65 years: 2.9%; *p* = 0.001). The level of education influenced the ratings for *around-the-clock availability of specialist care* (EL2: 47.1%, EL 3: 38.9%, EL1: 38.1%, EL4: 34.1%, EL5: 21.3%; *p =* 0.010), as well as for *hospice care* (EL5: 12.4%, EL2: 10.3%, EL3: 11.1%, EL4: 0.0%, EL1: 0.0%; *p =* 0.048), *psychological care* (EL4: 27.3%, EL5: 21.3%, EL3: 16.7%, EL2: 9.2%, EL1: 0.0%; *p =* 0.010) and *grief counseling* (EL1: 19.0%, EL4: 9.1%, EL3: 5.6%, EL2: 1.1%, EL5: 1.1%; *p =* 0.002). See Figure 2.

### 3.4. Survey Participants’ Views on the Fears of Palliative Care Patients

About one–third of participants (32.9%) said they believed the greatest fear of terminally ill people was that of *death*, followed by *pain* (24.1%), *losing independence* (16.9%), *being a burden to someone* (9.3%), *losing mental abilities* (7.6%), *becoming disabled* (6.0%), *loneliness* (1.0%) and *financial burdens* (0.5%). Age influenced expectations that fear of *death* was the greatest concern and was higher in participants in the group aged 35–49 years (47.5%) and lowest in the 65+ age group (11.3%; *p* < 0.000). Fear of *death* was chosen significantly more often by men than women (41.7% vs. 26.6%; *p =* 0.001).

A quarter of respondents (24.1%) believed the greatest fear was *pain,* but the results also depended considerably on age, with the youngest respondents (≤34 years) having the least fear of *pain* (15.8%) and respondents in the group aged 50–64 years having the greatest (32.0%) (*p* = 0.030). See Figure 3 and Table A3 in Appendix C for more details.

### 3.5. Survey Participants’ Views on the Best Place for Palliative Care Patients to Receive Care

When terminally ill, 39.0% of respondents believed it is best to be cared for *at home by professional caregivers*, followed by cared for *at home by relatives* (21.7%), *hospices* (21.4%) and *hospitals* (11.9%). Results depended strongly on sex, with females more often choosing to be cared for *at home by professional caregivers* (f: 40.3%, m: 36.5%) and *hospices* (f: 25.7%, m: 13.5%) and less often cared for *at home by relatives* (f: 20.4%, m: 24.0%) and *hospitals* (f: 7.3%, m: 20.2%; *p*= 0.005). Respondents with EL2 (44.8%), EL4 (47.7%) and EL 5 (37.1%) considered it best to be cared for *at home by professional caregivers*, while respondents that had EL 1 thought it best to be cared for *at home by relatives* (52.4%) and those that had EL 3 were in favor of *hospices* (31.5%) (*p =* 0.003). No differences were found between the various age groups (*p* = 0.533). 

### 3.6. Survey Participants´ Views on the Services That Should Be Provided to Palliative Care Patients and Their Relatives

The service that was mentioned most often was home pain management 69.9% (n = 293), followed by time off for family caregivers (58%), daycare (49.4%), nightcare (42.0%), home assistance (40.1%), information on financial support (35.1%), availability of social workers (27.4%), complementary and alternative medicine (22.4%) and pastoral care (20.5%). Sex differences in preferred services were only found in time off for family caregivers, which were significantly more popular among females (64.3%) than males (49.1%) (*p* = 0.002). Age groups significantly influenced preferences for complementary and alternative medicine (*p* = 0.001), pastoral care (*p* < 0.001), availability of social workers (*p* = 0.009) and information on financial support (*p* < 0.001). Educational level also had a significant influence on the preference for pastoral care (*p* < 0.001), availability of social workers (*p* = 0.004) and information on financial support (*p* = 0.012). See Figure 4 and Table A4 in Appendix C.

## 4. Discussion

With over 400 participants, this cross-sectional survey showed that in our sample, more people had heard of palliative care in Styria, Austria, than in many other countries [6]. In similar countries to Austria, a relatively large number of people (70%) had heard of palliative care but had no precise idea of what it really was [5,13]. Our survey also indicated that as few as 13.5% of respondents knew a fair amount or had extensive knowledge of it. It is interesting to note that there were major differences in terms of gender, with 40.6% of men saying they knew nothing about palliative care, as compared to only 21.7% of women. The difference may reflect gender roles, as informal caregiving is generally provided by women [14,15], which would explain why they knew more about it. At the same time, female caregivers report greater burden, stress, anxiety, depression, and adverse health outcomes than male caregivers [16]. A Canadian study found that the majority of spousal caregivers were female and that these female caregivers reported a significantly greater level of caregiving strain than their male counterparts [17]. In our survey, this was reflected in the question of what the respondents thought families and their relatives should be provided in terms of support. Here, significantly more women (64.3%) than men (49.1%) said that additional support should be offered to caregivers at home, so that caregiving relatives could have more time for themselves. It is evident that caring for relatives is still largely the responsibility of women [18] and that awareness of caregiver burden is therefore stronger in women. It has further been shown that care recipients with a female caregiver are less likely to receive informal support from family and friends in carrying out tasks associated with personal care than male caregivers, to whom others are more likely to offer such support [17].

Another field in which gender differences were observed was in response to the question where it is best to be provided palliative care when terminally ill. Women more often chose to be cared for at home by professional caregivers (f: 40.3%, m: 36.5%) or cared for in hospices (f: 25.7%, m: 13.5%) and less often to be cared for at home by relatives (f: 20.4%, m: 24.0%). This difference may also reflect the traditional distribution of roles, with women more frequently wanting to avoid burdening their families with their need for care. Another interesting aspect is that significantly more men (41.7%) than women (26.6%) said they thought the greatest fear of a terminally ill person was likely to be the fear of death. A survey conducted in Germany found that significantly more women than men found the thought of their own death less painful than that of a loved one [19]. 

However, differences were not only related to gender but also to age. Younger respondents tended to know less about palliative care, as has also been found in previous studies [20]. In the group aged 18–35 years, 10.1% of respondents believed that psychological support was the greatest need of terminally ill patients, whereas no respondents in the group aged 65 years and older thought so. For older people, around-the-clock availability of specialist care for relatives appeared to be more important than for younger people, while psychological support was more important to younger people than to older people. There were significant differences in opinions on the greatest fears of terminally ill patients. In the middle-aged group, 47.5% thought it was the fear of death, while only 11.3% believed this in the 65+ group. Respondents in the group up to 34 years old reported the least fear of pain (15.8%), while the greatest fear of pain was found in the group aged 50–64 years (32.0%).

Education was also a crucial factor influencing knowledge about palliative care. The more educated participants were, the more likely they were to know the main aim of palliative care. There were also education-related differences in answers to other questions, such as where terminally ill patients should best be cared for. More than 50% of respondents that had only completed compulsory schooling thought it best to be cared for at home by relatives, an opinion that was shared by only 37.1% of respondents with a university degree. There is also evidence in the literature that wealthier people and women were more likely to receive palliative care from specialists [21]. This attitude is also reflected in our survey, as it became apparent that the different groups of people had different levels of knowledge and different perceptions, fears and expectations with respect to palliative care. In an ageing society, in which more and more people will need looking after and require palliative care, targeted information is necessary. Furthermore, when providing such information, it is important to take into account that there are social group-related differences in knowledge about palliative care. In addition, age-, gender- and education-related differences exist in attitudes, fears and expectations with regard to palliative care and ageing in general.

Based on these results, it would make sense to avoid blanket information campaigns but rather tailor them according to the audience. For example, it is well known that people from socially disadvantaged population groups in Austria tend to gather information via social media [22]. As the results of this survey also confirm what we know about the distribution of health literacy, namely that people with little education, in financially precarious situations or who are looking for work tend to have limited health literacy [22], it is necessary to investigate the various possibilities to reach out to and inform these groups of people. A good example of targeted information and training in the field of palliative care for lay people are the Last Aid Courses, which are successfully held in various countries [23,24].

In order to support those that bear the greatest burden of care and thus to relieve society as a whole, information campaigns are required that target groups of people that know little about palliative care, so that they become more aware of, and take advantage of, available services. In this way, the quality of life of terminally ill patients and their caregivers in these groups may be increased. 

### Limitations of the Study

There may have been some bias in the sample, as the participants were recruited in medical practices and therefore do not include people that never consult a doctor. Another limitation is that the questionnaire was only distributed in a printed version and not as an online version and may therefore not have appealed to tech-savvy people, who may have a different level of awareness of palliative care than those in our sample and may know about different aspects of it. A limitation is also that all but one of the questions were closed-ended, and that only one question provided the opportunity to give an additional response over and above existing options. Thus, participants may have been unable to give the answers they would like. However, we were constrained in our ability to change the structure of the questionnaire by our interest in ensuring the international comparability of the results.

## 5. Conclusions

To facilitate the care of severely ill patients at home, it would make sense to develop targeted information campaigns in Western industrialized countries. This would also make it possible to reach out to groups, such as women, that bear the greatest burden of caregiving and to inform them of available support options. Such campaigns should also attempt to deliver information to less informed groups of people, such as young, poorly educated men, in order to raise their awareness of the difficulties and challenges of providing care to terminally ill patients.

## Figures and Tables

**Figure 1 healthcare-11-02611-f001:**
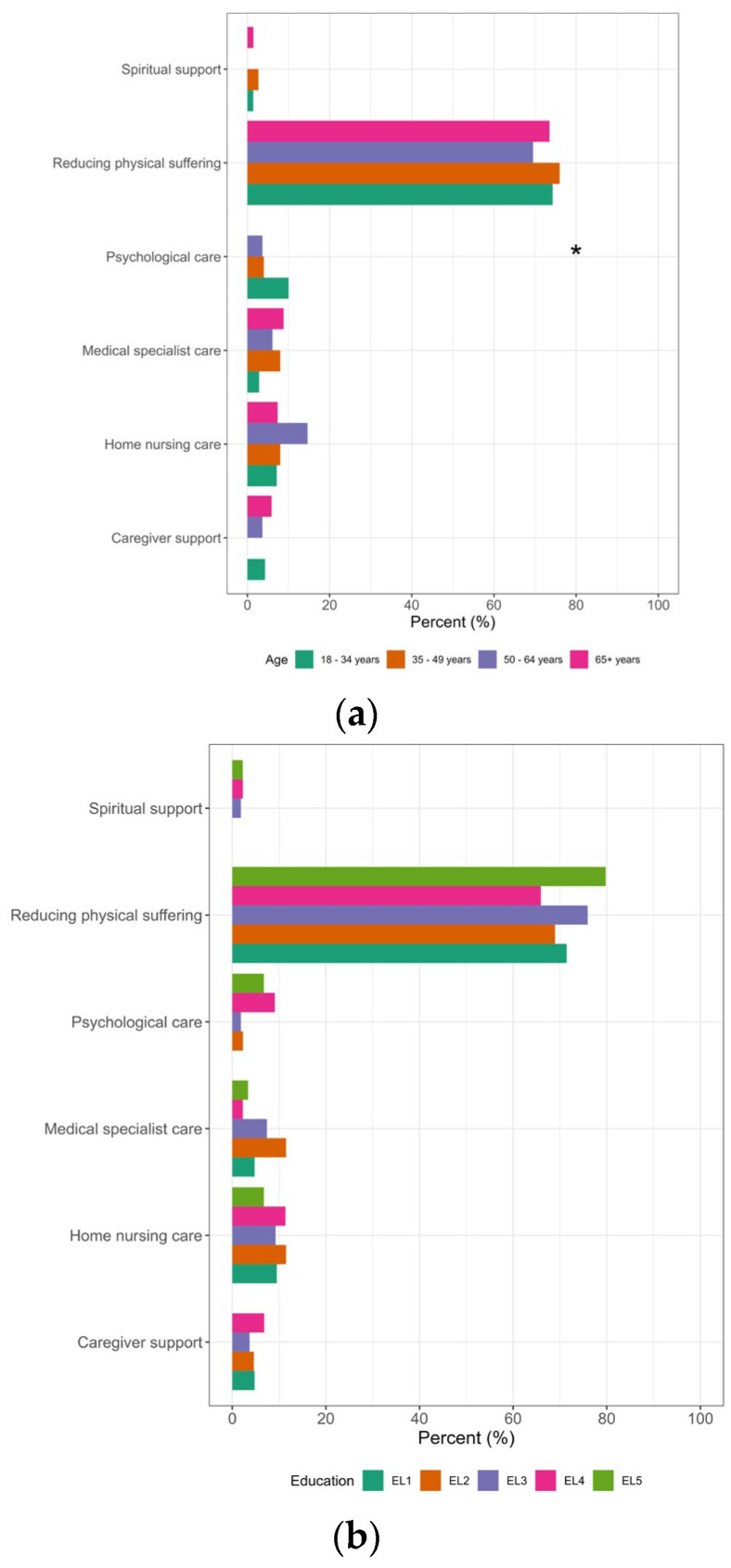
Differences in survey participants’ views on the greatest needs of palliative care patients according to the participating’s age, educational level (EL1: compulsory school, EL2: apprenticeship, EL3: trade school, EL4: high school, EL5: university) and sex. * Significant; (**a**) age; (**b**) education; (**c**) sex; n = 295.

**Figure 2 healthcare-11-02611-f002:**
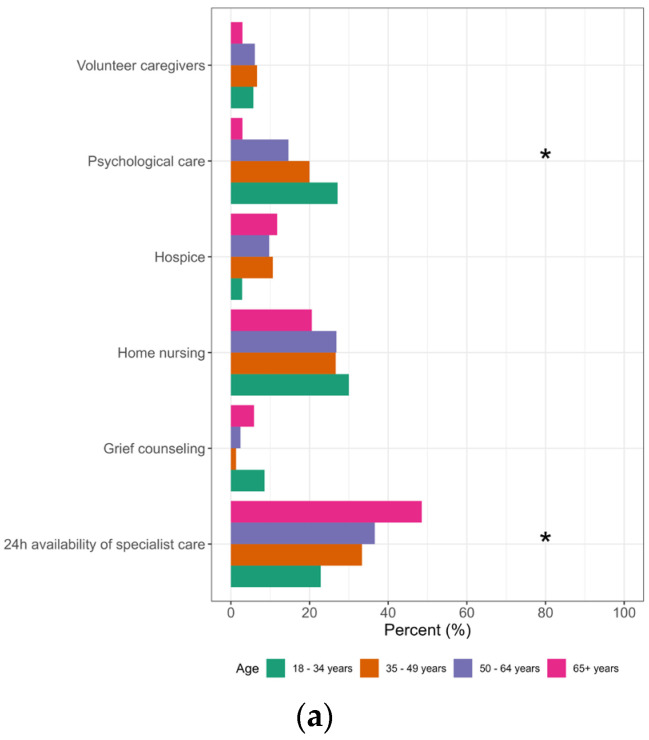
Differences in survey participants’ views on the greatest needs of their relatives according to the participant’s age, educational level (EL1: compulsory school, EL2: apprenticeship, EL3: trade school, EL4: high school, EL5: university) and sex. * Significant; (**a**) age; (**b**) education; (**c**) sex; n = 295.

**Figure 3 healthcare-11-02611-f003:**
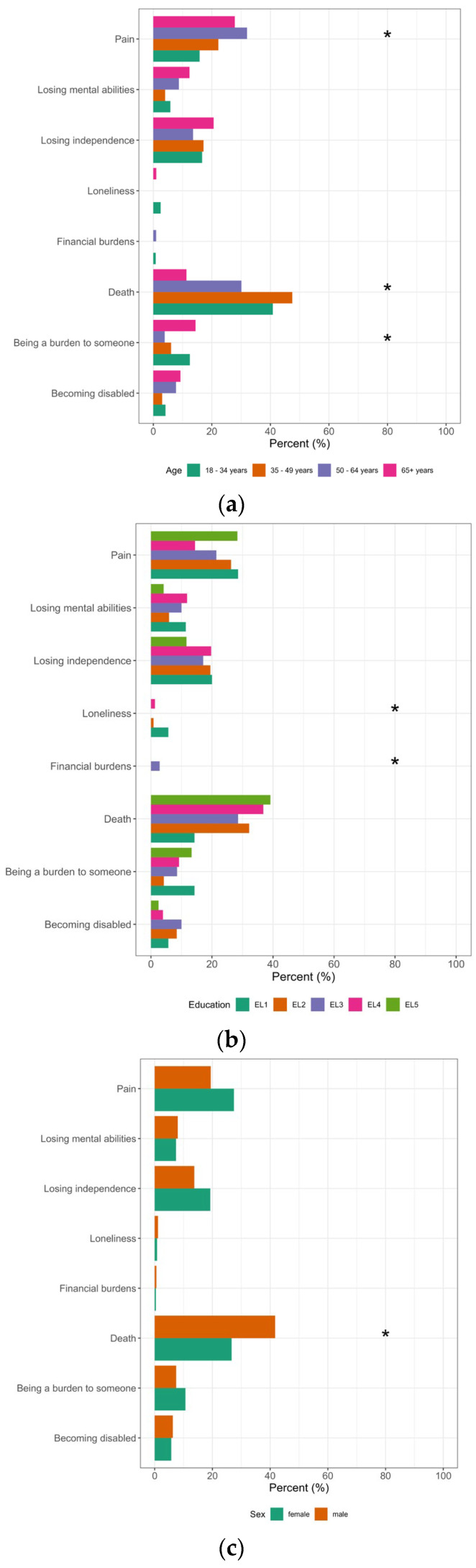
Differences in survey participants’ views on the fears of palliative care patients according to the participant’s age, educational level (EL1: compulsory school, EL2: apprenticeship, EL3: trade school, EL4: high school, EL5: university) and sex. * Significant; (**a**) age; (**b**) education; (**c**) sex; n = 419.

**Figure 4 healthcare-11-02611-f004:**
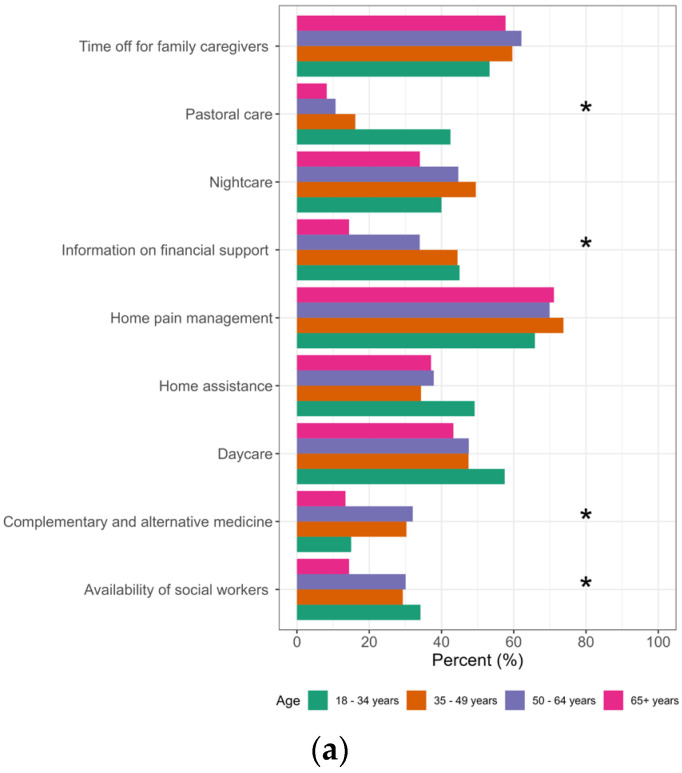
Differences in survey participants’ views on the services that should be provided to palliative care patients and their relatives according to the participating’s age, educational level (EL1: compulsory school, EL2: apprenticeship, EL3: trade school, EL4: high school, EL5: university) and sex. * Significant; (**a**) age; (**b**) education; (**c**) sex; n = 419.

## Data Availability

The datasets used and/or analyzed for the current study are available from the corresponding author on reasonable request.

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
