# Peer review of "Palliative Care Survey: Awareness, Knowledge and Views of the Styrian Population in Austria"

_healthcare, 2023, doi:10.3390/healthcare11192611_

Round 1
Reviewer 1 Report
The Manuscript presents and discusses the data from a survey aimed at understanding the perception of the population in Styria, Austria, regarding palliative care from both the patient’s and the caregiver’s perspectives. The study is of significant relevance in the literature on PC because it can provide interesting insights to define interventions that make the implementation of this model more systematic in the public health sector, identifying the reasons why patients struggle to accept it, notwithstanding its proven benefits. Moreover, studies like these are lacking in literature. I thank the Authors for the interesting reading.
However, there are a number of minor and major issues that in my opinion need to be addressed.
TITLE - Title is way too long, I suggest to make it shorter.
ABSTRACT - There are several typos and errors that impact the perceived quality of the work. For example:
Line 19: “(…) to identify THEIR potential needs (…)” (ITS, if it refers to POPULATION).
Line 25: “(…) aged 50 to <65.” The symbol < is useless here.
Line 27: “73,2%” should be 73.2%.
Line 31: “The best place to be when terminally ill” do Authors mean the best place to die? The sentence can be misunderstood.
I also suggest avoiding including methodological details such as the software used for statistical analysis in the abstract (see lines 22-23).
INTRODUCTION - In my opinion, the introduction has two main limitations: the first concerns the literature, which is often lacking, and the second relates to the fact that there is some confusion between palliative care and early palliative care.
1. Regarding the literature, I recommend supporting certain key statements in the manuscript with bibliographic references. For example, the very first sentence (lines 46-47) should be supported by some bibliographic references. There are several studies that attest to the benefits of palliative care, depending on whether the authors are referring to standard palliative care or early palliative care (see the following comment). The statement in lines 64-65 should also be supported by the literature (see, for example, Alcade & Zimmermann, 2022; Bandieri et al., 2023; Shen & Wellman, 2019; Zimmermann, 2019).
As I mentioned earlier, this study is valuable not only for its interest but also because there are few studies of this kind published in the literature or shared by organizations/societies/foundations. Due to their scarcity, I believe it's worthwhile to cite all – or almost all – of them, even briefly. Therefore, I recommend including in the bibliography:
- Center to Advance Palliative Care. New Public Opinion Research Reveals Palliative Care Still Relatively Unknown among the General Public: Shows Education for Consumers and Physicians Necessary to Make a Difference. Available online: https://www.capc.org/about/press[1]media/press-releases/2019-8-8/new-public-opinion-research-reveals-palliative-care-still-relatively-unknown-among-general[1]public-shows-education-consumers-and-physicians-necessary-make-difference/
- Claxton-Oldfield, S.; Claxton-Oldfield, J.; Rishchynski, G. Understanding of the Term “Palliative Care”: A Canadian Survey. Am. J. Hosp. Palliat. Care 2004, 21, 105–110. https://doi.org/10.1177/104990910402100207.
- Wallace, J. Scottish Partnership for Palliative Care Public Awareness of Palliative Care: Report of the Findings of the First National Survey in Scotland into Public Knowledge and Understanding of Palliative Care; Scottish Partnership for Palliative Care: Edinburgh, UK, 2003; ISBN 978-0-9542396-4-0.
- Bin Ahmed, I.A.; Bin Ahmed, A.A.; Bin Ahmed, I.A.; AlFouzan, S.K. A Descriptive Online Survey about the Knowledge of Palliative Care Residents of Saudi Arabia Has Compared to the General Worldwide Population. SMJ 2020, 41, 537–541. https://doi.org/10.15537/smj.2020.5.25062.
- Community Attitudes to Palliative Care Issues, Rural Health and Palliative Care; Australian Department of Health and Ageing, Canberra. Publications Production Unit, Governance and Business Strategy Branch: Canberra, Australia, 2003
2. Regarding the lack of clarity between palliative care and early palliative care, these are two different models. The term "palliative care" usually refers to the original model of standard, late-stage palliative care, while the term "early palliative care" refers to a more modern approach of care initiated within 8 weeks of diagnosis, for patients with a prognosis of 6-24 months. The two paradigms have different goals and provide different interventions. Upon reading the Manuscript and especially the cited literature, it's not clear to me whether the Authors are interested in standard late-stage palliative care, early palliative care, or palliative care in a broad sense (regardless of timing of intervention). For example, references 1 and 3, to name a couple, pertain to early palliative care, yet the terms "early" or "timely" are not mentioned anywhere in the Manuscript.
In an era where early PC is proving to be more effective than standard PC, I believe it's necessary for the literature to clarify its stance on this matter. Therefore, I suggest that the Authors mention the distinction between the two models in the introduction, even briefly, and explain if the study aimed to investigate the perception of the term "palliative care" as the historical, late model, as the more recent, early paradigm, or in a general sense, without delving into the referral timing aspect.
3. Regarding reference 6, I believe it's not accurate to state that 32% of the sample knew nothing about palliative care. This would imply a neutral and, all things considered, favorable situation, as the general public is more open to new ideas when starting from a blank slate. In fact, the Turkish survey reveals a much worse situation, i.e., that 32% of the population believes PC to be something different from what it actually is. Please, correct me if I’m wrong. Otherwise, I think data from cited studies should be reported more consistently.
4. I apologize to the Authors, but to be honest, I was not familiar with the term "Styria" before now. It might be helpful to add a couple of words referring to the fact that it is a (beautiful) Austrian region.
5. I recommend rephrasing the study's aims as they are currently unclear. I believe they could be clearer if the reader is informed that the survey asked participants to imagine themselves as patients receiving palliative care or as family members of patients on palliative care. It would also be helpful to explicitly state that the sample consisted of healthy individuals.
6. Please, correct typos like that in line 71 (“futher” instead of “further”).
MATERIALS AND METHODS - I suggest organizing the Materials and Methods section in a more systematic manner. Currently, there are scattered pieces of information about the procedure in different subsections. Additionally, the procedure itself is described somewhat vaguely. While reading it, I had many doubts that, in my opinion, should be clarified in the text.
1. Firstly, I wouldn't assume that the reader is aware of why Austrian participants were required to know German. One might expect them to need knowledge of Austrian, but I assume that German is the official language in Styria, correct? Were the participants required to be native speakers, or could they know German as a second language?
2. Was the survey multiple-choice? Could participants provide multiple answers, or was the choice forced?
3. It's not clear why two general practitioners recruited participants. So, were the participants patients, or were they healthy individuals? If they were healthy individuals visiting their doctor, do you think that recruiting only sick people, even if not suffering from conditions necessarily terminal or requiring palliative care, could have biased the responses?
4. If the survey was anonymous, how was it delivered to maintain anonymity?
5. Please, explain that in some questions participants had to imagine to be patients and in other questions they had to imagine to be familiars.
6. There are some concepts that are not clear to me, and I apologize for this, but they might not be clear to the reader as well. Particularly, from lines 92-100. What do the Authors mean when they say they used a questionnaire that was used SUCCESSFULLY on Slovenian adults? What do the Authors mean when they say the questionnaire was BACK-TRANSLATED by a native Slovenian speaker? Was the initial translation by the professional translator proofread by a native Slovenian speaker working in the medical field?
7. What is the Cognitive Debriefing method? It would be helpful to have a reference and a couple of lines of explanation.
8. I appreciate that the questionnaire was made available as supplementary material; however, the fact that it's not in English makes it rather useless. I recommend providing an English translation as well. For instance, I cannot ascertain whether the questionnaire inquired whether the participant had any close family member or friend with an oncological or onco-hematological illness; this could have potentially biased their responses.
9. Was it specified to empathize with terminally ill patients or only with patients receiving palliative care?
RESULTS
1. The labels in Figure 1 are very small and not very readable; I recommend increasing their size font.
2. The sentence "Views on the greatest needs of terminally ill persons depended strongly on the respondent's age" (lines 142-143) is somewhat overstated, considering there's a significant difference in only one of the 6 proposed needs. I believe it's generally appropriate to explicitly mention in the text, beyond the figure, that there are no other significant relationships between the given responses and the variables of age, education, and gender.
3. From line 159 to line 165, and subsequently in other parts of the Results section, I suggest replacing the levels of education with the abbreviations introduced earlier, EL1, EL2, EL3, etc.
DISCUSSION
1. Authors were talking about Styria, but in the Discussion they talk about Austria. Do they think that their results can be generalized to all the Austrian population? In the Introduction they state that “Styrian institutions specializing in the care of patients with life limiting illnesses have existed since 1998” (lines 68-69). Does this apply to other Austrian regions as well?
I appreciate the effort to explain and contextualize the data, which, although speculative, provide interesting points for reflection. However, it's challenging for me to evaluate the discussion without being able to read the survey questions. And this, in my opinion, is the main limitation of the manuscript, which can be easily addressed by incorporating a translation. In general, it appears to me that the survey is not so much focused on the perception of palliative care, but rather on the perception of illness in its terminal phase. This shifts the focus of the study, although it continues to be helpful in implementing interventions to spread awareness of palliative care as well as improving the paradigm itself.
Another doubt I have is how the response options given to participants were arrived at. Did the Authors consider if there were other responses that participants might have wanted to provide but were not offered by the survey? Did the questions include the option "OTHER - SPECIFY"?
Therefore, I reserve the right to reevaluate the study when a translation of the survey is made available in English.
I recommend proofreading the text.
Reviewer 2 Report
Thank you for the opportunity to review this manuscript. The findings no doubt will have great utility to increasing public awareness around the use of palliative care. I offer the following suggestions to strengthen this manuscript as follows:
Abstract: Lines 33-35: This sentence says "this item was chosen by women significantly more often by women" This seems circular.
The literature review is significantly lacking and thus the significance of the study is not clear to this reviewer. The following questions should be addressed by the literature:
1) The authors note the differences by gender and education based on their findings and literature to that effect is in the discussion section but not mentioned in the literature review.
2) Significance-Why is this study needed? What is the connection between awareness and service usage? Is palliative care underutilized in this region?
3) The authors cite an existing measure from a Solvenian study in the methods section-what were the findings from that study specifically? That study should be mentioned in the literature review.
This reviewer recommends adding a clearly stated research question prior to the methods section.
Recruitment and data collection: You say "two general practices" please clarify are these healthcare practices?
The graphs provided are difficult to read-this review suggestions a table format for more extensive data reporting with potentially the selection of some key graphs for the publication.
Discussion section: Suggest a revision to the first sentence "With other 400 participants, this cross-sectional survey showed that more people had heard of palliative care in Austria than in many other countries." The current wording implies that a representative sample was used such that the findings can be applied across Austria. The authors should revise to make it clear that the findings apply to this Austrian sample and not all Austrians.
A limitations section should be added that identifies the study limitations, including the use of a convenience sample which may introduce some level of bias.
Survey instrument: This reviewer recommends including an English-translated copy of the survey.
The abstract should be reviewed for English language in particular. No other major issues noted.
Reviewer 3 Report
Thank you for your work!
I apprecheated a lot the attempt to raising awareness about such an important topic which is crucial in this time. Due to an increase in the spread of palliative care facilities and opportunities, also with reference to new specialisation schools for doctors, this paper is a good contribution.
One note to pose refers to the fact that the study is a sort of simulation, because it is referred to persons who were asked "to imagine that they were palliative care patients or family members of palliative care patients". But I understand both the difficulty to refer directly to persons involved in PC, and that also a simulation reports the right feelings.
In line 88 probably is "Data were...";
The figures are very important, but too small. It is quite difficult in fact to follow the descriptions of the results along the text, so having figures more readible would actually help the readers.
I would suggest:
1) to put in evidence, probably in another work, the sensibility to the spiritual aspects of PC; it is crucial in reference to the basic and decisive foundation of PC by C. Saunders and E. Kubler Ross;
2) to propose an English version of the full questionaire, not only to facilitate the understanding, bur essentially to offer the possibility to other scholars to propose similar studies in other countries with the same basic instrument to be translated, after the Slovenian and German (probably with an increased reference to the Spiritual aspect, as said) for a larger and interconnected study across countries!
In any case I consider the paper a good work.
Thank you!
Round 2
Reviewer 2 Report
The revisions provided greatly improve the manuscript. I recommend the following minor change:
For the figures provided, please identify the number of respondents for each graph as indicated by "n ="
English quality is adequate
Author Response
Thank you very much for taking the time to review our manuscript again.
Comments and Suggestions for Authors: The revisions provided greatly improve the manuscript. I recommend the following minor change: For the figures provided, please identify the number of respondents for each graph as indicated by "n =" Thank you for the input. We have added the number of respondents for each graph.